# Evaluating the Effect of Iron(III) in the Preparation of a Conductive Porous Composite Using a Biomass Waste-Based Starch Template

**DOI:** 10.3390/polym15112560

**Published:** 2023-06-02

**Authors:** Laria Rodríguez-Quesada, Karla Ramírez-Sánchez, Sebastián León-Carvajal, Giovanni Sáenz-Arce, Fabián Vásquez-Sancho, Esteban Avendaño-Soto, Juan José Montero-Rodríguez, Ricardo Starbird-Perez

**Affiliations:** 1Master Program in Medical Devices Engineering, Instituto Tecnológico de Costa Rica, Cartago 159-7050, Costa Rica; 2Centro de Investigación en Servicios Químicos y Microbiológicos (CEQIATEC), Escuela de Química, Instituto Tecnológico de Costa Rica, Cartago 159-7050, Costa Rica; 3Departamento de Física, Facultad de Ciencias Exactas y Naturales, Universidad Nacional, Heredia 86-3000, Costa Rica; 4Centro de Investigación en Óptica y Nanofísica, Departamento de Física, Universidad de Murcia, 30100 Murcia, Spain; 5Materials Research Science and Engineering Center (CICIMA), University of Costa Rica, San José 11501-2060, Costa Rica; 6School of Physics, University of Costa Rica, San José 11501-2060, Costa Rica; 7Escuela de Ingeniería Electrónica, Instituto Tecnológico de Costa Rica, Cartago 159-7050, Costa Rica

**Keywords:** biopolymer, porous material, starch, conductive composites, biomass waste

## Abstract

In this work, the effect of iron(III) in the preparation of a conductive porous composite using a biomass waste-based starch template was evaluated. Biopolymers are obtained from natural sources, for instance, starch from potato waste, and its conversion into value-added products is highly significant in a circular economy. The biomass starch-based conductive cryogel was polymerized via chemical oxidation of 3,4-ethylenedioxythiophene (EDOT) using iron(III) p-toluenesulfonate as a strategy to functionalize porous biopolymers. Thermal, spectrophotometric, physical, and chemical properties of the starch template, starch/iron(III), and the conductive polymer composites were evaluated. The impedance data of the conductive polymer deposited onto the starch template confirmed that at a longer soaking time, the electrical performance of the composite was improved, slightly modifying its microstructure. The functionalization of porous cryogels and aerogels using polysaccharides as raw materials is of great interest for applications in electronic, environmental, and biological fields.

## 1. Introduction

Natural polymers enable the production of innovative materials within the framework of the principles of the circular economy, and it is a key strategy to balance progress and sustainability [1]. The circular economy concept is based on material resource reuse, reducing the generation of waste, relying on efficient green technologies that reduce the use of hazardous chemicals, and promoting growth through innovation [2]. A biomass is a sustainable, green, non-toxic [3], biodegradable [4], and abundant natural resource [5], and a significant portion of lignocellulosic biomass waste is discarded from agricultural activities [6]. For instance, starch is a natural polymer obtained from potato agricultural biowastes [7,8]. The management of biomass waste is a growing challenge in developing countries due to its increasing generation and the lack of feasible and efficient approaches to provide proper use of these resources [9]. Their conversion into value-added products is possible through their structuration and functionalization [10]. Accordingly, our approach opens a novel use of starch biomass to prepare added-value composites using green technologies.

From the technological perspective, the capacity to modulate biomaterial properties to convey unique characteristics allows their application in different fields. Biopolymers are excellent raw material candidates for aerogel processing regarding circular economy principles, relying on renewable raw materials or energy sources [5,11,12]. Aerogels and cryogels are solid, lightweight, and highly specific surface area materials with interconnected networks of particles obtained from a wet gel during a process where their liquid phase is removed and replaced with gas without collapsing the solid structure [13,14]. Specifically, cryogels are obtained by freeze-drying hydrogels at sub-zero temperatures, and the solvent (i.e., water) is thereafter sublimated from the solid state directly into the vapor state [15,16]. Freeze-drying is reported as a simple, environmentally friendly, and economical technique for producing highly porous materials with reduced shrinkage [17]. The obtained cryogels using this technique are porous solid materials with large pores and high inner surfaces [11,18].

Conductive porous materials possess a unique combination of properties and have become a global hot topic in the fields of catalysis, drug delivery, electronic devices, extracellular matrices, and tissue regeneration [11,19,20,21,22,23,24,25]. Biopolymer-based conducting porous materials have been previously obtained through the chemical oxidation of 3,4-ethylenedioxythiophene (EDOT) using iron(III) ions [14,19,21,26]. Poly 3,4-ethylenedioxythiophene (PEDOT) is an intrinsically conductive polymer that has been used commercially in electrolytic capacitors and as anti-static coatings [27,28,29]. The electrical conductivity and responsiveness of the ionic conductor depend on the final network structure of the porous composite and on the strategies used to create the conductive material [14,19,21,26]. The polymer is obtained by electrodeposition, oxidative molecular layer deposition [30], and oxidative chemical vapor deposition [31] methods that provide high conductive and conformal films with thickness control. Several oxidants are used during the chemical deposition PEDOT synthesis, including persulfate [32], copper(II) [33], and iron(III) (widely reported chloride and tosylate) [30,34,35]. Highly conductive PEDOT:tosylate films have been successfully obtained via vapor phase deposition for cell cultures under electrical stimulation, only limited by a small amount of polymer deposited [23]. In the absence of a template, morphology and molecular structure are strongly affected by the oxidant/monomer ratio [36]. At a molecular level, it is mainly affected by the formation of charge carriers [24], and at the macro scale, film thickness and crystallinity influence the bulk polymer conductive properties [30]. Specifically, the interaction of the oxidant agent (i.e., iron(III) ions) with polysaccharides (e.g., starch, dextran, and cellulose) has been widely reported as coordinated to the oxygen donor atoms interaction [37,38,39]. Since modification of biopolymers may depend on the synthesis methodology, it is of our interest to address the interaction effect of the oxidant and the biopolymer template interaction time in the properties of the porous conductive composite.

In this work, the effect of iron(III) in the preparation of a conductive porous composite using a biomass waste-based starch template was evaluated. The starch-based conducting porous material was obtained through the chemical oxidation of 3,4-ethylenedioxythiophene (EDOT) using iron(III) p-toluenesulfonate. Our study is a step forward to previous works [14,19,21] since our porous template was produced from biomass waste, and the oxidant-template interaction showed to be pivotal in the electrical performance of the conductive composite. The physical properties were evaluated on polymerized cryogels by nitrogen physisorption analysis, pore size distribution through scanning electron microscopy (SEM), and gas pycnometer studies. Finally, the study of the cryogel composition after polymerization was evaluated via Fourier-transformed infrared spectroscopy (FTIR), thermogravimetric analysis (TGA), an X-ray diffractometer (XRD), total organic carbon (TOC) studies, and energy dispersive spectrometry analysis (EDS).

## 2. Materials and Methods

### 2.1. Materials and Reagents

Sodium hydroxide (ACS, 99% purity), hydrochloric acid (ACS, 99% purity), 3,4-ethylenedioxythiophene (EDOT, 97% purity), iron(III) p-toluenesulfonate hexahydrate (quality level 100), potassium bromide (KBr, FTIR grade), and 2-Propanol (IPA, ACS reagent, quality level 300) were purchased from Sigma-Aldrich (San José, Costa Rica).

### 2.2. Starch Isolation from Agro-Industrial Waste Potatoes

Potato tubers from agricultural waste were obtained from a local market in Cartago, Costa Rica, and were used as biomass source in accordance with CONAGEBIO permission R-CM-ITCR-002-2023-OT. The biomass was proceeded based on the following references [40,41].

Briefly, starch isolation was carried from the tubers using ca. 300 g of discarded potatoes, then thoroughly washed using deionized water and detergent, peeled, and sliced into small pieces. The tuber pieces were ground using a home blender (Hamilton Beach, Glen Allen, VA, USA), and the homogenate obtained product was suspended in 1 L of deionized water containing NaOH (0.1% wt.). The suspension was filtered, and sample was allowed to settle down for 4 h.

The obtained sediment was washed four times using NaOH (0.1% wt.), and it was neutralized using HCl, and the supernatant was removed. The obtained residue was centrifuged for 5 min at 5000 rpm and stored at −20 °C for 24 h. Then, the sample was lyophilized for 72 h at −60 °C (Martin Christ, Alpha 1-2LD plus, Harz, Germany) and stored in conical tubes before their use. The starch extraction yield percentage was calculated through a mass analysis of the fresh tuber and the amount of starch obtained after lyophilization.

Once the starch was isolated, starch-based cryogels were prepared. First, an aqueous solution containing starch 9% wt. was autoclaved (Tomin, TM-322, New Taipei City, Taiwan) for 7 min at 121 °C and 1 atm. Then, samples were maintained for 48 h at 4 °C for starch retrogradation [11]. The resulting sample was freeze-dried via lyophilization for 72 h at −60 °C (Martin Christ, Alpha 1-2LD plus).

### 2.3. Starch Cryogels Polymerization

Once the cryogels were obtained, PEDOT was synthesized via oxidative chemical polymerization of EDOT onto the starch scaffold, adapted from previous work [14]. The starch cryogel sample was placed in an alcoholic solution containing isopropanol (IPA) and iron(III) p-toluenesulfonate hexahydrate 0.3 M. The effect of the oxidizing agent on the synthesis of PEDOT was evaluated by soaking the starch cryogel in an iron(III) solution at different times (0.5, 8, 24, and 48 h). Then, the samples were rinsed 5 times using IPA. Cryogels samples were immersed for 48 h in an IPA solution containing EDOT (2 M), resulting in a dark blue composite. The samples were then washed with IPA until the solution was colorless. The resulting scaffold was dried for 24 h in a vacuum oven (ADP 200C, Yamato-Scientific, Tokyo, Japan) at 45 °C and 85 kPa.

### 2.4. Physical Characterization of the Porous Materials during the Functionalization Process

Skeletal density of polysaccharide-based cryogels (𝜌_𝑠𝑘𝑒𝑙_) was determined using a nitrogen pycnometer (Ultrapyc 5000, Anton Paar, Graz, Austria) set at room temperature and 15 psi. A total of 10 replicates were used for each measurement. The bulk density (𝜌_𝑏𝑢𝑙𝑘_) of the cylindrical cryogels was calculated by weighing and measuring the dimensions of each cryogel before functionalization and after soak times in iron(III) and EDOT polymerization. Equations (1) and (2) were used to calculate cryogels’ overall percentage porosity (ε) and total pore volume (𝑉𝑝), respectively, according to a previous report [42].
(1)$$\varepsilon \;=1-\frac{\rho_{bulk}}{\rho_{skel}}\;\times 100$$
(2)$$Vp=\frac{1}{\rho_{bulk}}-\frac{1}{\rho_{skel}}$$

The specific surface area (A_BET_) of the polymerized cryogels at different soak times in iron(III) solution was determined by applying nitrogen physisorption using the BET (Brunauer-Emmett-Teller) method in an adsorption analyzer (Anton Paar, Autosorb iQ, Graz, Austria). The characterization of the specific pore volume (*Vp*) and the mean of pore diameter (Dp) were estimated using the BJH (Barrett-Joyner-Halenda) method. Finally, micrographs of the polymerized cryogels were recorded via scanning electron microscopy (JSM-IT500 InTouch Scope; JEOL, Tokyo, Japan).

### 2.5. Analysis of the Composition of the PEDOT Polymerized in Porous Materials after Different Soak Times in Iron(III)

The composition of the cryogels was analyzed using a Fourier-transformed infrared spectroscopy (FTIR) spectrometer (Nicolet 380, Thermo Scientific, Madison, WI, USA) controlled with the OMNIC v9.3.30 software. For the analysis, PEDOT-polymerized starch cryogels were ground using a mortar and mixed with potassium bromide (KBr). The obtained powder was pressed using a Specac Atlas™ manual 15T hydraulic press (Specac Ltd., Orpington, UK) to obtain a pellet. Measurements of the acquired pellets were made in transmittance mode in the 4000–400 cm^−1^ spectral range using 64 scans with a resolution of 4 cm^−1^.

Thermogravimetric analysis (TGA) was carried out for all cryogel samples after iron(III) immersion and polymerization in an SDT Q600 from TA Instruments (New Castle, DE, USA). A nitrogen atmosphere (100 mL·min^−1^) was maintained during the analysis with a scan rate of 10 °C·min^−1^ from 25 °C to 800 °C in alumina cups (110 μL) (TA Instruments, New Castle, DE, USA). Finally, the atmosphere was changed to synthetic air at a flow of 100.00 mL·min^−1^.

The crystal structure of the inorganic components was studied with the X-ray diffractometer (XRD) (Empyrean, Malvern-PANalytical, Malvern, UK). The bulk material was packed on a sample holder and measured in the range of (2θ) 5°–55° with a cobalt (Co) anode and an Fe filter using a GaliPix detector with a 17.8 mm anti-scatter slit. The operating conditions including step size, voltage, and current were adjusted to 0.007°, 40 kV, and 40 mA, respectively. Finally, the 2θ data was recalculated from Cobalt-K_α_ to Copper-K_α_ for analysis, the background was removed, and each diffractogram was normalized by area.

The total organic carbon (TOC) for the porous composites was measured using a CHNS analyzer (Vario TOC cube, Elementar, Langenselbold, Germany) using ca. 50 mg of sample. Scanning electron microscopy (SEM) and energy-dispersive X-ray spectroscopy (EDS) analysis for the St/iron(III)/PEDOT cryogels after different soak times in iron(III) solution were analyzed using an energy dispersive X-ray spectrometer detector (EDS) (JSM-IT500, JEOL, Tokyo, Japan) and tabletop SEM (CUBE II, Emcrafts, Gwangju, Republic of Korea) with EDS Xplore Compact, Oxford, UK), applying a voltage of 30 kV. Image analysis was performed with the SMILE VIEW™ Lab software 2.1 (JEOL Ltd., Tokyo, Japan).

### 2.6. Effect of the Starch–Porous Material Iron(III) Soak Time on the Electrical Properties of the PEDOT Polymerized Cryogels

The electrical properties of the conductive material were evaluated via electrical impedance spectroscopy on a potentiostat instrument (AUTOLAB, PGSTAT-302 model, Utrecht, The Netherlands). The analysis was carried out in a system of two electrodes made of copper strips; the conductive material was located between the 2 copper electrodes placed in a compression load of ca. 35 mg. The resistivity of each probe was obtained from the current vs. voltage curves corrected by the area and length. The impedance analysis was performed at 50 mV in a frequency range of 0.1 to 10^5^ Hertz (Hz).

## 3. Results and Discussion

### 3.1. Starch Extraction from Biomass Waste and Its Structuration

The obtained starch from the discarded potato matter was in the range of 7.82 ± 0.10%, in agreement with those values reported in the literature [43]. The extracted starch was used to generate a starch-based cryogel from a retrogradation of an aqueous solution and freeze-drying process (some properties are shown in the Appendix A). The obtained porous materials presented similar physical properties to those previously fabricated from commercial starch [21] (see Figure 1), exhibiting a novel strategy to process biomass waste into value-added products.

### 3.2. Characterization of the Starch Template, Iron(III)/Starch, and Conductive Composite Cryogels

The chemical composition of the composites at different soaking times was evaluated via infrared, thermogravimetry, and elemental composition. The obtained data confirmed starch-iron interaction and its effect on the conductive polymer deposition in the porous structure. Starch cryogels were poured into the IPA-iron(III) solution, and immediately, the coloration of the samples changed from white to dark yellow. This is attributed to the iron content in the solution, supposing that free ferric ions in the IPA-iron(III) solution bonded to the polysaccharide structure in accordance with previous reports [44].

The modification of the starch cryogel sample caused by the iron ions was monitored via TGA analysis. As is shown in Figure 2, the degradation mechanism of the cryogel samples was modified by the presence of iron ions. At lower soaking times (0.5 h), the material degrades, following a similar trend to that of the single template. However, longer soaking times show a decrease in the starch thermal stability, and no significant changes were observed after 8 h [45].

FTIR spectra of starch cryogels and PEDOT-polymerized starch cryogels after IPA-iron(III) soaking are shown in Figure 3. The main characteristic absorption peaks of the polymerized samples in the region of 3000–3500 cm^−1^ show a shape change and shifting in the functionalized samples (Figure 2b,c), indicating a strong interaction between the polysaccharide structure and iron(III) ions [14,37,44]. Specifically, the absorption peaks at wavelengths around 3300 cm^−1^ could be attributed to the O–H stretching vibrations of structural starch OH. Signal changes between samples could imply the participation of starch hydroxyl groups in the complexion reaction [44,46]. Additionally, the signal at 2900 cm^−1^ corresponding to the C–H stretching vibration presented slight intensity differences among samples.

The signal at 1640 cm^−1^ is indicative of the stretching vibration absorption peak of C=O [37,46]. Shifting of this signal when the conductive polymer is deposited on the template may be due to changes in the starch/PEDOT interaction (see Figure 3 inset). The main bands found in the FTIR spectra at 1125, 1078, 1039, and near the region of 1024 cm^−1^ were attributed to coupled valent vibrations of the C–O and C–C bonds and deformational vibrations of the C–C–H and C–O–H bonds [46].

The band at 1153 cm^−1^ is identified for the starch template, and it was assigned to vibrations of the C–O–C group on the glycosidic bond; this signal shifted to 1125 cm^−1^ in polymerized samples, possibly related to the polysaccharide–iron interaction [47].

The signals at 860 and 692 cm^−1^ are identified in the starch/iron template and the polymerized sample (Figure 3b,c). Those bands are absent in the starch template spectrum. The same result was obtained previously by some researchers who mentioned that these signals correspond to the β-FeOOH group, confirming the formation of the polysaccharide-iron(III) complexes [47,48,49].

Finally, the signal at 1438 cm^−1^ is associated with thiophene symmetric C=C stretching and its oxidation state, confirming the polymerization of EDOT in all samples [50] (see Appendix A, in Appendix A).

The textural properties of the resulting cryogels for each synthesis stage are shown in Appendix A (see in Appendix A). All samples presented low bulk densities (𝜌_𝑏𝑢𝑙𝑘_) with differences among St, St/iron(III), and St/iron(III)/PEDOT materials. These differences are attributable to the presence of iron(III) and PEDOT, as well as the reduction of empty spaces in the cryogels. The results agreed with previous reports using a similar concentration of corn starch [51] and for PEDOT three-dimensional structures [26]. 

Differences were determined between skeletal density values. St and St/ iron(III) cryogels showed skeletal densities (𝜌*_skel_*) values in the range of native starch (1.50 g·cm^−3^) [16]. The pycnometer results showed that the St/iron(III)/PEDOT skeletal density decreased significantly, reaching similar values to those previously reported for PEDOT cryogels [26,52]. Porosity was determined using Equation (1), which is directly related to the geometrical scaffold density. The obtained values were higher than 85% for all samples.

We observed that the cryogel macroporous structure endured the polymerization process through all preparation stages without an evident collapse, as shown in Figure 4, even at long soak times. Regarding the microstructure, the nitrogen absorption data indicated low specific surface areas and micropore size for all the samples, which is common in samples processed in freeze-drying conditions [31]. Samples tended to show lower surface areas and a slight decrease in the micropore size at longer soak times. The physical properties of the obtained porous materials are summarized in Appendix A (see in Appendix A).

TGA data demonstrated that no significant difference in ashes was observed after an 8 h soaking time, and a similar amount of conductive polymer was observed in the 24 h and 48 h of soak time samples (see Table 1). Consequently, EDS analysis confirmed that iron is present in the samples, and sulfur (associated with the PEDOT polymer) (Figure 5) was deposited onto the template (see overlayed images).

XRD diffraction (Figure 6) analysis showed patterns for the starch template in all the samples [53]. After the PEDOT deposition, a peak profile associated with the conductive polymer is observed, showing some shifting due to molecular structural rearrangements caused by changes in its doing level [35,54]. Additionally, small, broad, and overlayed peaks are observed in the XRD data; they could be related to amorphous iron compounds. Similar results were reported during the synthesis of PEDOT using FeCl_3_ as an oxidant in the absence of a template [34]. Therefore, considering the iron and PEDOT tendency to saturate the template after an 8 h soak time, our FTIR and XDR results may indicate that iron ions differ in their coordination interactions with the starch template along the soak time.

Elemental analysis data (Appendix A, see in Appendix A) showed that the ratio of sulfur in the sample decreased after 8 h of soaking. The difference between TOC and TGA results suggests that a longer soaking time decreases the amount of counter ion (i.e., p-toluenesulfonate ion) in the matrix. The counter ion is allowed to freely diffuse to the solvent since a longer interaction time results in an effective stabilization via coordination of the iron species into the starch template [38].

Electrical impedance spectroscopy characterization (EIS) of the porous composites was performed using a previous model [21], which included the mass transfer phenomena, the effects of the distribution of energy states, the porosity of the materials, and the electrical conductivity. The impedance magnitude and phase of porous composites at different soaking times are shown in Figure 7. The 0.5 h soaking time sample shows a low impedance and a resistive behavior, which is probably due to the existence of free ions (e.g., iron(III) ions) [55,56]. A longer soak time unveiled higher impedance values and more complex electrical behavior.

The real impedance (Z′) and imaginary impedance (Z″) (Figure 8) confirmed the effect of the starch/iron interaction time in the electron transfer mechanism on the porous conductive samples (Figure 7). At a short iron starch interaction time (0.5 h), a dominant resistive behavior was identified. Zhang et. al. have reported that the concentration of iron(III) ions in a polymeric network decreases the electrical resistance [56]. Longer soaking times unveiled a more complex electrical mechanism, including contact resistance, the impedance linked to the intraparticle porous material, and the impedance related to the mass transfer region as well as a second semicircle, suggesting a complementary electrical mechanism due to the iron complexation by the starch. In general, the data describe a system containing two layers or two regions with different electrical properties [57].

The impedance data fitting was performed using Autolab NOVA 2.1.6. based on a previous model [21]. The model consists of an equivalent electrical circuit comprising a series connection of two parallel circuits. The parameters obtained from the impedance modeling are reported in Appendix A (see in Appendix A). The 0.5 and 8 h samples were not modeled using the complete model because these samples showed a single semicircle. Longer soaking times unveiled higher impedance values and more complex electrical behavior because of the lower availability of free iron ions by complexation for the biopolymer matrix along with the diffusion of counter ions during the sample preparation. The fitting data confirmed that a longer soaking time led to a higher capacitance for the first and second semicircles, which is expected for conductive polymer deposition [21].

Overall, biomass-waste starch was a suitable raw material to structure a porous template. The porous structure withstands the fabrication process with only slight differences in microporosity. The chemical composition at different soaking times was studied via infrared, thermogravimetry, and elemental composition, confirming that a longer soak time altered the starch–iron ions interaction. Additionally, the results evidenced the presence of PEDOT and counter ions through the synthesis process, and they influenced the electrical behavior of the obtained composites.

## 4. Conclusions

Our work gives insights into the methodology to functionalize starch obtained from biomass waste using chemical polymerization of poly(3,4-ethylenedioxythiophene) directly on the substrate. Thermal and spectrometric characterization of those composites allow us to detect and quantify the iron–starch interaction and the amount of oxidizing agent during the soaking time. Moreover, we have confirmed using electronic microscopy and nitrogen physisorption that the porous template endured the polymerization process through all preparation stages without an evident collapse on the macrostructure, slightly affecting the microstructure due to the PEDOT deposition on the template surface. Longer soaking times (i.e., 48 h) unveiled, via impedance spectroscopy, a more complex electrical behavior, which is of great interest to a wide audience, including electronic, environmental, and biological applications. Finally, the conversion of potato waste into a conductive porous composite is an example of the potential preparation of innovative materials that balance progress and sustainability in a circular economy framework. 

## Figures and Tables

**Figure 1 polymers-15-02560-f001:**
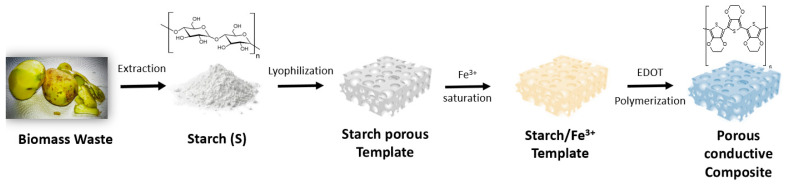
Starch processing scheme from biomass to final conductive structured composite.

**Figure 2 polymers-15-02560-f002:**
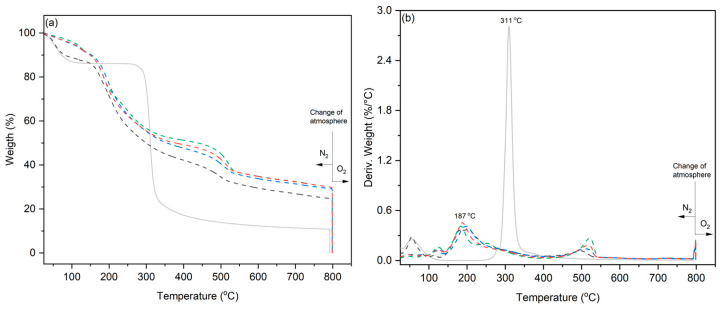
Thermogravimetric (**a**) and derivate (**b**) analysis at different soaking times in IPA-iron(III) solution of porous materials at different soak times for (solid grey line) 0, (dashed black line) 0.5, (dashed green line) 8, (dashed blue line) 24, and (dashed red line) 48 h.

**Figure 3 polymers-15-02560-f003:**
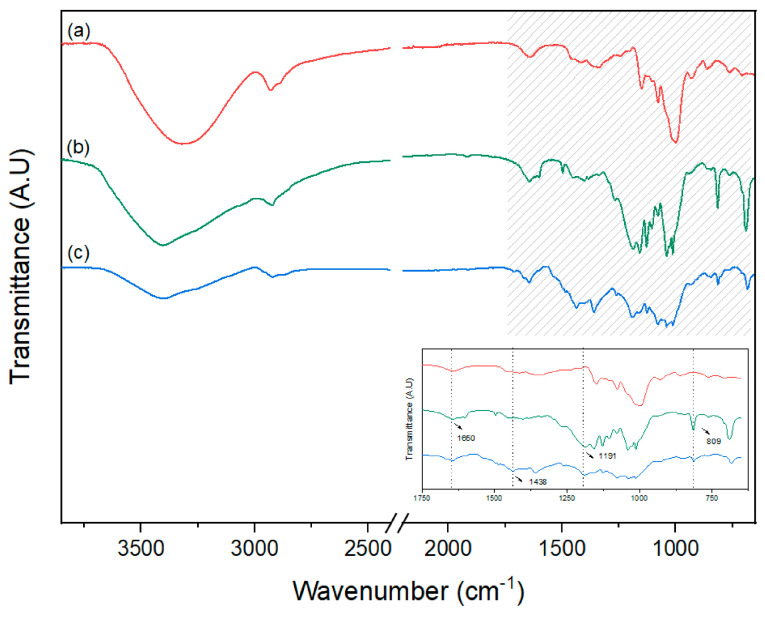
Analysis via infrared spectroscopy of (**a**) starch template, (**b**) starch/iron(III) composite, and (**c**) the polymerized porous material. Inset shows the zoom-in for 1650, 1438, and 809 cm^−1^ signals.

**Figure 4 polymers-15-02560-f004:**
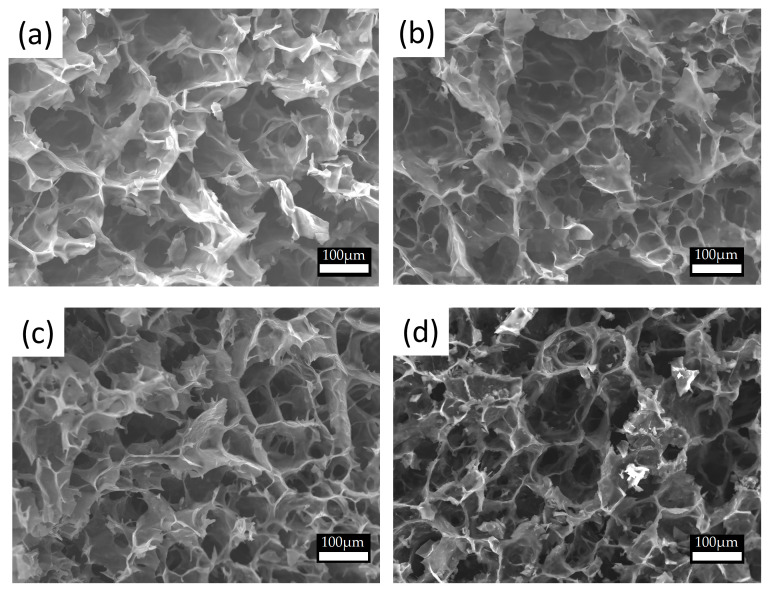
Macrostructure for different porous materials after the synthesis of PEDOT at different soak times in iron(III) solution for (**a**) 0.5, (**b**) 8, (**c**) 24, and (**d**) 48 h obtained via SEM. Scale bar 100 μm.

**Figure 5 polymers-15-02560-f005:**
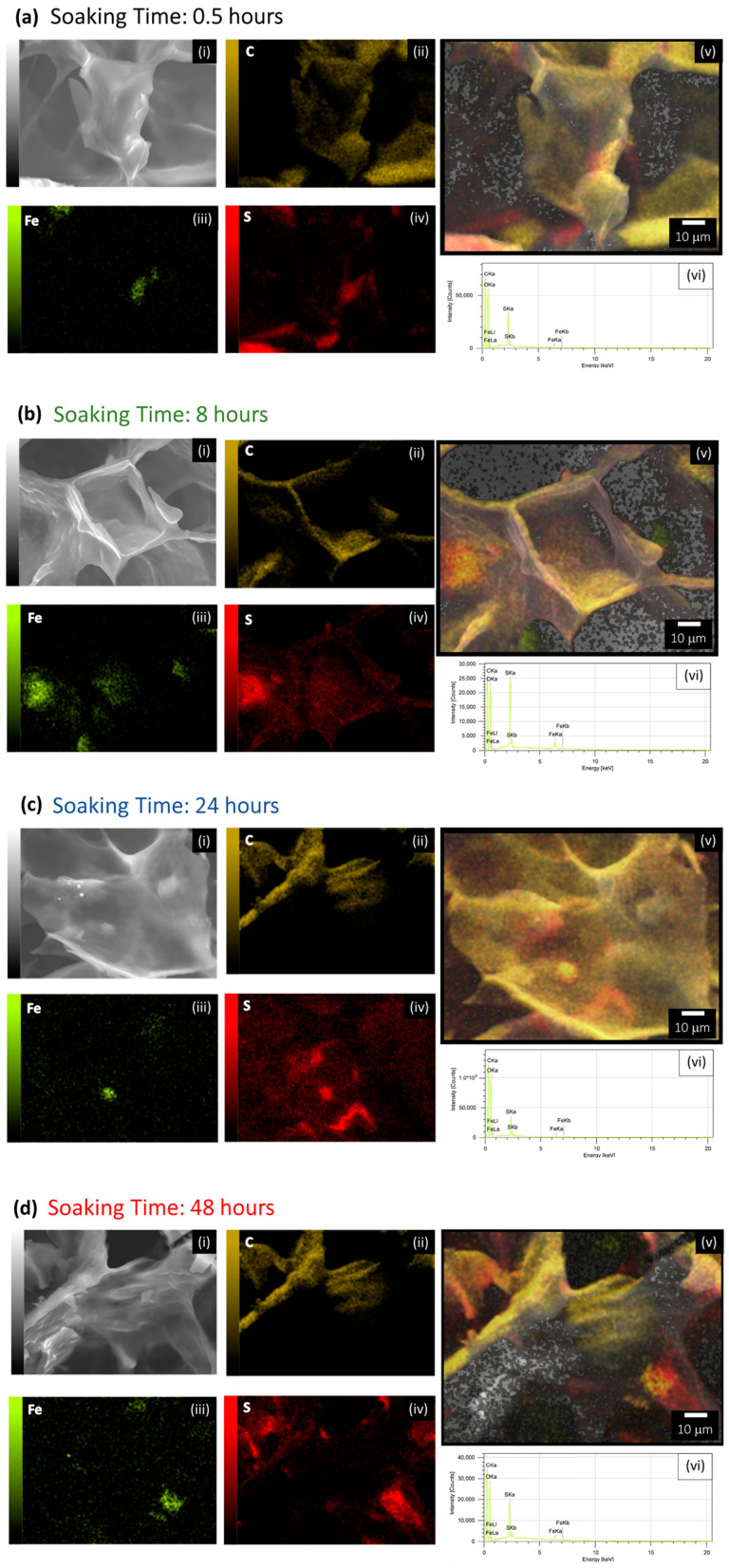
Scanning electron microscopy-energy dispersive spectrum (SEM-EDS) analysis of the porous material at different soak times in iron(III) solution for (**a**) 0.5, (**b**) 8, (**c**) 24, and (**d**) 48 h. For all the samples, (i) SEM images, (ii) carbon atom mapping, (iii) iron atom mapping, (iv) sulfur atom mapping, (v) overlayed images, and (vi) energy-dispersive spectroscopy (EDS) spectra.

**Figure 6 polymers-15-02560-f006:**
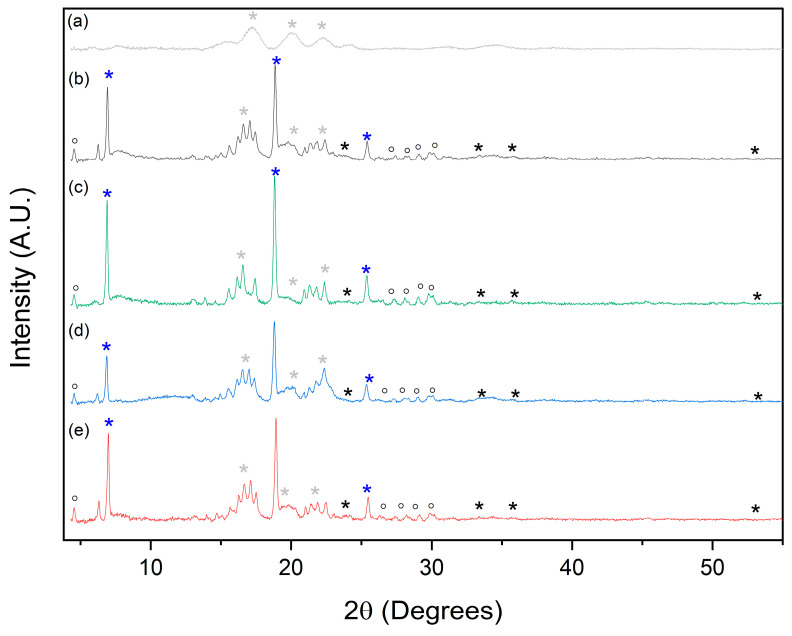
XRD diffractograms of (**a**) the starch template and the conductive composites samples obtained after (**b**) 0.5, (**c**) 8, (**d**) 24, and (**e**) 48 h soak time in Fe^3+^ solution. Symbols for signals related to (*) starch template, (*) PEDOT polymer, and (◦, *) iron compounds.

**Figure 7 polymers-15-02560-f007:**
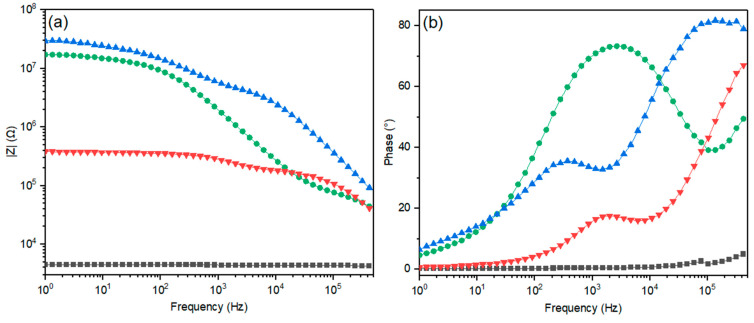
Comparative impedance measurements at 50 mV (**a**) Bode and (**b**) phase plots for the conductive porous materials obtained at different soaking times in iron(III) solution. Legend: (black squares) 0.5, (green circles) 8, (light blue triangles) 24, and (red triangles) 48 h.

**Figure 8 polymers-15-02560-f008:**
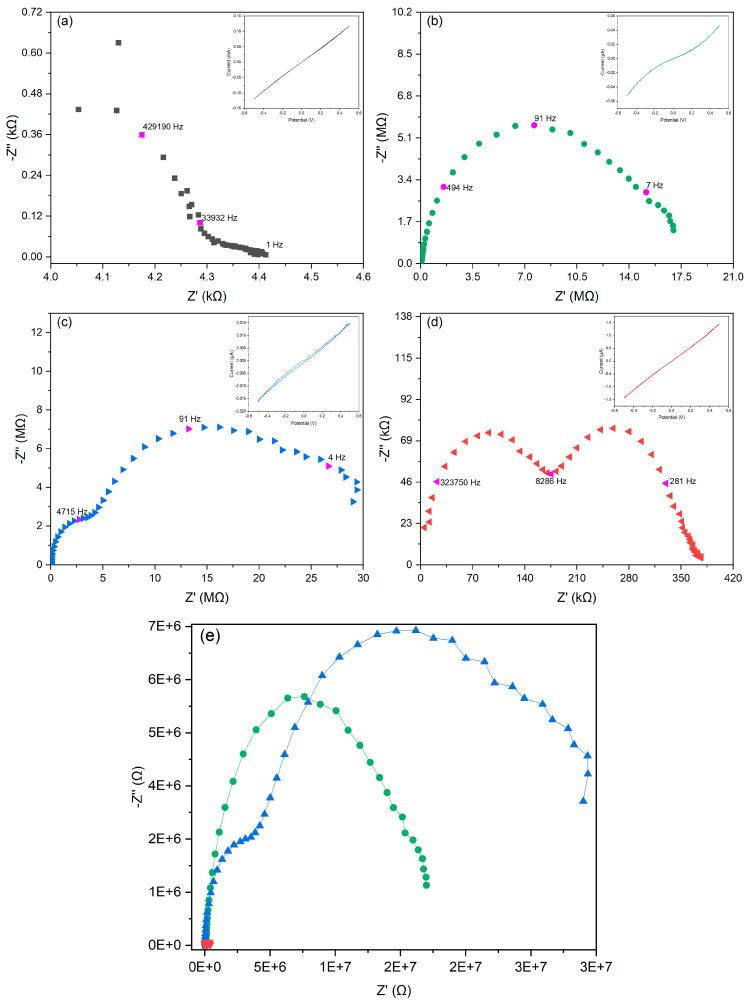
Individual Nyquist plots for (**a**) 0.5 (black squares), (**b**) 8 (green circles), (**c**) 24 (light blue triangles), and (**d**) 48 h (red triangles) samples. Arbitrary frequencies are remarked in pink. Inset shows cyclic voltammetry (CV) curve for each sample. (**e**) Comparative Nyquist plot of obtained conductive composite cryogels at different soaking times in iron(III) solution determined using electrochemical impedance spectroscopy at 50 mV.

**Table 1 polymers-15-02560-t001:** PEDOT obtained in the structure of porous materials at different soak times in iron(III).

Conductive Cryogels at Different Soak Time in Iron(III) Solution (h)	Sample Ashes (%)		
	Iron(III) Absorption	Total Residues Obtained after PEDOT Synthesis	PEDOT Obtained in the Structure
0.5	24.24	27.13	2.89
8	28.22	34.04	5.82
24	29.12	35.70	6.58
48	29.57	36.12	6.55

## Data Availability

The raw data of this study are available from the corresponding authors (L.R.-Q. and R.S.-P) on request.

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
