# Peer review of "Evaluating the Effect of Iron(III) in the Preparation of a Conductive Porous Composite Using a Biomass Waste-Based Starch Template"

_polymers, 2023, doi:10.3390/polym15112560_

Round 1
Reviewer 1 Report
1. The novelty statement of the manuscript is completely missing in the introduction
2. There shall be one section to include all the chemicals (purity, brand, and company) and materials.
3. The author should provide further information about biomass waste
4. The quality of all the figures should be improved
5. The visible scale bar should be provided
6. The discussion section should be expanded to highlight the scientific contribution of this study to this field.
7. In the results need rearrange in the sequence of importance.
8. The conclusion should be revised to be informative and show only significant findings of the study.
9. What are possible technology-oriented applications of the work for commercialization purposes?
10. How would this research work advance the previous work done in the existing field of study and/or across other fields?
11. Future work and Social implications shall be highlighted in the conclusion
Author Response
Manuscript ID: polymers-2291667, Research Article.
Title: "Preparation of a conductive porous composite via iron(III) chemical oxidative deposition using on a biopolymer based -cryogel produced from biomass waste"
Authors: Laria Rodríguez-Quesada¹*, Karla Ramírez-Sánchez², Sebastián León-Carvajal1, Giovanni Sáenz-Arce3-4, Fabián Vásquez-Sancho5-6, Esteban Avendaño-Soto5-6, Juan José Montero-Rodríguez7 and Ricardo Starbird-Perez²
We are very grateful to all the reviewers for their helpful and thoughtful comments. We addressed all the concerns and modified the text accordingly. A detailed response to each comment is provided below:
Reviewers' Comments to Author:
Reviewer: 1
COMMENTS FOR THE AUTHORS
- “The novelty statement of the manuscript is completely missing in the introduction”
RESPONSE: Thank you for your valuable observations, we have modified the introduction section according to your comment.
Main modifications could be observed from line 35 to line 56: “Natural polymers enable the production of innovative materials within the framework of the principles of the circular economy and it is a key strategy to balance progress and sustainability [1]. Biomass is a sustainable, green, non-toxic [2], biodegradable [3] and abundant natural resource [4]. A significant portion of lignocellulosic biomass waste is discarded from agricultural activities [5]. Starch from potato agricultural production is an example of a natural polymer obtained from biowaste [6,7]. The management of biomass waste is a growing challenge in developing countries due to its increasing generation and the lack of feasible and efficient approaches to provide proper use of these resources [8]. Their conversion into value-added products is possible through its structuration and functionalization.
From the technological perspective, the capacity to modulate the biomaterial properties to convey unique characteristics allows their application in different fields. Biopolymers are excellent raw material candidates for aerogel processing regarding circular economy principles, relying on renewable raw material or energy sources [4,9,10]. Aerogels and cryogels are solid, lightweight, and high specific surface area materials with interconnected networks of particles obtained from a wet gel during a process where their liquid phase is removed and replaced with gas, without collapsing the solid structure [11,12]. Specifically, cryogels are obtained by freeze drying hydrogels at low temperatures, and the solvent (i.e. water) is thereafter sublimated from solid state directly into the vapor state [13]. Freeze drying is reported as a simple, environmentally friendly and economic technique for producing highly porous materials with reduced shrinkage [14]. The obtained cryogels using this technique are porous solid materials with large pores and high inner surfaces [9,15].”
- “There shall be one section to include all the chemicals (purity, brand, and company) and materials.”
RESPONSE: We appreciate your constructive comment. Specifications were included for all reagents.
From line 97 to line 100: “Sodium hydroxide (ACS, 99% purity), Hydrochloric acid (ACS, 99% purity), 3,4-ethylenedioxythiophene (EDOT, 97% purity), iron(III) p-toluenesulfonate hexahydrate (quality level 100), potassium bromide (KBr, FTIR grade) and 2-Propanol (IPA, ACS reagent, quality level 300) were purchased from Sigma-Aldrich (San José, Costa Rica).”
- “The author should provide further information about biomass waste”
RESPONSE: We appreciate your comment. We have provided more information related with biomass waste on the introduction (line 35 to line 44) and methodology sections (line 102 to line 105).
From line 35 to line 44: “Natural polymers enable the production of innovative materials within the framework of the principles of the circular economy and it is a key strategy to balance progress and sustainability [1]. Biomass is a sustainable, green, non-toxic [2], biodegradable [3] and abundant natural resource [4]. A significant portion of lignocellulosic biomass waste is discarded from agricultural activities [5]. Starch from potato agricultural production is an example of a natural polymer obtained from biowaste [6,7]. The management of biomass waste is a growing challenge in developing countries due to its increasing generation and the lack of feasible and efficient approaches to provide proper use of these resources [8]. Their conversion into value-added products is possible through its structuration and functionalization”
From line 102 to line 105: “Potato tubers obtained from agricultural waste were obtained from a local market in Cartago, Costa Rica, and was used as biomass source in accordance with CONAGEBIO permission R-CM-ITCR-002-2023-OT. The biomass was proceeded based on the following references [36,37].”
- The quality of all the figures should be improved
RESPONSE: Thank you for your observation. We have reviewed the images quality and confirmed that they are satisfactory according to the criteria of the Polymers journal
- The visible scale bar should be provided.
RESPONSE: We appreciate your observation. The scale bars have been corrected to improve their visibility in the revised manuscript.
- The discussion section should be expanded to highlight the scientific contribution of this study to this field.
RESPONSE: Thank you for your observation. The discussion section was corrected according to your recommendation to highlight the scientific contribution of the use of biomass waste during porous structures fabrication and conductive materials production influenced by soak times of porous materials in iron, in order to use it for several applications.
From line 328 to line 334: “Overall, biomass-waste starch was a suitable raw material to structure a porous template. The porous structure withstands all the fabrication process with only slightly differences in the microporosity. The chemical composition at different soaking times was studied by infrared, thermogravimetry and elemental composition, confirming that longer soaking times influence the starch thermal stability. Additionally, the results evidenced the presence of iron and PEDOT through the synthesis process, and those substances showed particularly effect on the electrical behavior of the obtained composites.”
- In the results need rearrange in the sequence of importance.
RESPONSE: We appreciate your thoughtful recommendation. Our results have been organized to confirm the starch-iron interaction, followed by the conducive composite characterization. An alternative arrangement may affect our manuscript outline.
- Thank you for your recommendation. We have improved the conclusion according to your observation, emphasizing the innovation all over the manuscript.
RESPONSE: Thank you for your recommendation. We have improved the conclusion according to your observation, emphasizing the innovation all over the manuscript.
From line 337 to line 350: “Our work gives insights in the methodology to functionalize starch obtained from biomass waste using chemical polymerization of the poly(3,4-ethylenedioxythiophene) directly on the substrate. Thermal and spectrometric characterization of those composites allows us to detect and quantify the iron-starch interaction and the amount of oxidizing agent along the soaking time. Moreover, we have confirmed using electronic microcopy and nitrogen physisorption that the porous template endured the polymerization process through all preparation stages without an evident collapse on the macrostructure, slightly affecting the microstructure due to the PEDOT deposition on the template surface. Longer soaking times (i.e., 48 hours) unveiled, by impedance spectroscopy, a more complex electrical behavior, which is of great interest to a wide audience, including electronic, environmental, and biological applications. Finally, the conversion of potato waste into a conductive porous composite is an example of the potential preparation of innovative materials that balance progress and sustainability in a circular economy framework.”
- What are possible technology-oriented applications of the work for commercialization purposes?
RESPONSE: The details of the technology-oriented applications related to the system proposed on this manuscript have been added to the introduction section and as conclusion.
Additional references were added to the document to support the potential use of functionalize porous materials: [Liu, Y., Liu, J., & Song, P. (2021). Recent advances in polysaccharide-based carbon aerogels for environmental remediation and sustainable energy. Sustainable Materials and Technologies, 27, e00240, Alvarado-Hidalgo, F., Ramírez-Sánchez, K., & Starbird-Perez, R. Molecules, 25(22), and F. Quignard, F., Valentin, R., & Di Renzo, F. (2008). Aerogel materials from marine polysaccharides. New journal of chemistry, 32(8), 1300-1310].
From line 45 to line 48: “From the technological perspective, the capacity to modulate the biomaterial properties to convey unique characteristics allows their application in different fields. Biopolymers are excellent raw material candidates for aerogel processing regarding circular economy principles, relying on renewable raw material or energy sources [4,9,10].”
- How would this research work advance the previous work done in the existing field of study and/or across other fields?
RESPONSE: Thank you for your recommendation. The main significance of our approach in the current manuscript have been reviewed following your observation.
From line 86 to line 89: “Our study is a step forward to previous works [12,16,18], since our porous template was produced from biomass waste and the oxidant-template interaction showed to be pivotal on the electrical performance of the conductive composite”
- Future work and Social implications shall be highlighted in the conclusion
RESPONSE: Thank you for your recommendation. We have improved the conclusion according to your observation, emphasizing the innovation of the manuscript.
From line 344 to line 349: “Longer soaking times (i.e., 48 hours) unveiled, by impedance spectroscopy, a more complex electrical behavior, which is of great interest to a wide audience, including electronic, environmental, and biological applications. Finally, the conversion of potato waste into a conductive porous composite is an example of the potential preparation of innovative materials that balance progress and sustainability in a circular economy framework.”
Reviewer 2 Report
Laria Rodríguez-Quesada et al. reported “Preparation of a conductive porous composite via iron(III) chemical oxidative deposition using on a biopolymer based-cryogel produced from biomass waste”. The work is publishable after addressing the following issues.
1. The title is not appealing for the readers. Make it comprehensive and short.
2. Introduction need extensive revision. Cryogel is mention in the title and I have find any very less discussion about cryogels in the introduction part. Discuss cryogels literature more in detail in the introduction part. Read and cite the following papers about cryogels which will be very helpful to improve the introduction part. Hybrid cryogels composed of P(NIPAM-co-AMPS) and metal nanoparticles for rapid reduction of p-nitrophenol Polymer 193 (2020) 122352, Highly porous cryogels loaded with bimetallic nanoparticles as an efficient antimicrobial agent and catalyst for rapid reduction of water-soluble organic contaminants Journal of Environmental Chemical Engineering 9 (2021) 106510.
3. How you say that its cryogel? Although you prepared the sample at 4 ˚C. What is the definition of cryogel? Clear this point or change the word from cryogels to hydrogels throughout the manuscript.
4. Ferric ions in the IPA-iron(III) solution 181 interacted and were bound to the polysaccharide structure. Explain this interaction with detail in the revised manuscript.
5. XRD analysis should be included in the revise manuscript for the confirmation of iron nanoparticles inside the network.
6. Where is the supporting information? EDS mapping should be included along with elemental table in the main manuscript.
7. Figure 6 is not clear. Make it clear in the revise manuscript.
8. Conclusion need improvement.
9. Check the grammatical and spelling mistake throughout the manuscript.
Author Response
Manuscript ID: polymers-2291667, Research Article.
Title: "Preparation of a conductive porous composite via iron(III) chemical oxidative deposition using on a biopolymer based -cryogel produced from biomass waste"
Authors: Laria Rodríguez-Quesada¹*, Karla Ramírez-Sánchez², Sebastián León-Carvajal1, Giovanni Sáenz-Arce3-4, Fabián Vásquez-Sancho5-6, Esteban Avendaño-Soto5-6, Juan José Montero-Rodríguez7 and Ricardo Starbird-Perez²
We are very grateful to all the reviewers for their helpful and thoughtful comments. We addressed all the concerns and modified the text accordingly. A detailed response to each comment is provided below:
Reviewer: 2
COMMENTS FOR THE AUTHORS
- The title is not appealing for the readers. Make it comprehensive and short.
RESPONSE: Thanks for the suggestion. We have corrected the title accordingly to your recommendation. Please: “Evaluating the effect of iron(III) in the preparation of a conductive porous composite using a biomass waste-based starch template”.
- Introductions need extensive revision. Cryogel is mentioned in the title and I have find any very less discussion about cryogels in the introduction part. Discuss cryogels literature more in detail in the introduction part. Read and cite the following papers about cryogels which will be very helpful to improve the introduction part. Hybrid cryogels composed of P(NIPAM-co-AMPS) and metal nanoparticles for rapid reduction of p-nitrophenol Polymer 193 (2020) 122352, Highly porous cryogels loaded with bimetallic nanoparticles as an efficient antimicrobial agent and catalyst for rapid reduction of water-soluble organic contaminants Journal of Environmental Chemical Engineering 9 (2021) 106510.
RESPONSE: Thank you very much for your observation. We have reviewed the introduction section in our manuscript including definitions and background regarding the application and cryogels basic concepts.
- How you say that its cryogel? Although you prepared the sample at 4 ˚C. What is the definition of cryogel? Clear this point or change the word from cryogels to hydrogels throughout the manuscript.
RESPONSE: We appreciate your valuable comment. Regarding to the cryogel definition, it has been discussed in previous research [Alvarado-Hidalgo, F., Ramírez-Sánchez, K., & Starbird-Perez, R. Molecules, 25(22)]. Cryogels are porous materials obtained after the process of freeze drying of hydrogels. Information related with definition and properties of cryogels have been added to the introduction section.
- Ferric ions in the IPA-iron(III) solution 181 interacted and were bound to the polysaccharide structure. Explain this interaction with detail in the revised manuscript.
RESPONSE: Thank you for your recommendation. As stated based on the references in the manuscript, carbohydrates can form strong interactions between oxygen atoms present in their structures and the metal ions, However, another reference: [Gyurcsik, B., & Nagy, L. (2000). Coordination chemistry reviews, 203(1), 81-149] has been added to the manuscript to support the interaction process between biopolymer and metal.
- XRD analysis should be included in the revise manuscript for the confirmation of iron nanoparticles inside the network.
RESPONSE: We appreciate your kind recommendation. However, we did not expect iron nanoparticles in our material. XDR confirmed different iron compounds in the final composites (please see supplementary information), and likely in further investigations there could be focus on the preparation of nanoparticles, but it is out of our current scope.
- Where is the supporting information? EDS mapping should be included along with elemental table in the main manuscript.
RESPONSE: Thank you for your observation. We have added the elemental data on the supplementary information as figure S2 and mentioned in the revised manuscript.
- Figure 6 is not clear. Make it clear in the revise manuscript.
RESPONSE: Thank you for your observation. We have changed the figure 6 according with your observation.
- Conclusion need improvement.
RESPONSE: Thank you for your recommendation. We have improved the conclusion according to your observation, emphasizing the innovation of the manuscript.
From line 337 to line 349: “Our work gives insights in the methodology to functionalize starch obtained from biomass waste using chemical polymerization of the poly(3,4-ethylenedioxythiophene) directly on the substrate. Thermal and spectrometric characterization of those composites allows us to detect and quantify the iron-starch interaction and the amount of oxidizing agent along the soaking time. Moreover, we have confirmed using electronic microcopy and nitrogen physisorption that the porous template endured the polymerization process through all preparation stages without an evident collapse on the macrostructure, slightly affecting the microstructure due to the PEDOT deposition on the template surface. Long soak time (i.e., 48 hours) unveiled, by impedance spectroscopy, a more complex electrical behavior, which is of great interest to a wide audience, including electronic, environmental, and biological applications. Finally, the conversion of potato waste into a conductive porous composite is an example of the potential preparation of innovative materials that balance progress and sustainability in a circular economy framework.”
- Conclusion need improvement Check the grammatical and spelling mistake throughout the manuscript.
RESPONSE: Thank you for the observation. We have fully reviewed the manuscript and have corrected the grammatical and spelling errors.
Reviewer 3 Report
The manuscript authored by Ricardo Starbird-Perez and co-workers reports the fabrication of conductive aerogels through the functionalization of naked porous starch aerogels by the polymerization of 3,4-ethylenedioxythiophene (EDOT). I consider that the work lacks novelty to be published in Polymers because the synthesis and characterization of starch/PEDOT aerogels have already been reported by other authors (DOI: 10.1016/j.msec.2013.12.032; 10.1016/j.carbpol.2018.02.040). In addition to this, the tests carried out are very simple, and no additional information is provided concerning previous works.
Author Response
Manuscript ID: polymers-2291667, Research Article.
Title: "Preparation of a conductive porous composite via iron(III) chemical oxidative deposition using on a biopolymer based -cryogel produced from biomass waste"
Authors: Laria Rodríguez-Quesada¹*, Karla Ramírez-Sánchez², Sebastián León-Carvajal1, Giovanni Sáenz-Arce3-4, Fabián Vásquez-Sancho5-6, Esteban Avendaño-Soto5-6, Juan José Montero-Rodríguez7 and Ricardo Starbird-Perez²
We are very grateful to all the reviewers for their helpful and thoughtful comments. We addressed all the concerns and modified the text accordingly. A detailed response to each comment is provided below:
Reviewer: 3
COMMENTS FOR THE AUTHORS
- The manuscript authored by Ricardo Starbird-Perez and co-workers reports the fabrication of conductive aerogels through the functionalization of naked porous starch aerogels by the polymerization of 3,4-ethylenedioxythiophene (EDOT). I consider that the work lacks novelty to be published in Polymers because the synthesis and characterization of starch/PEDOT aerogels have already been reported by other authors (DOI: 10.1016/j.msec.2013.12.032; 10.1016/j.carbpol.2018.02.040). In addition to this, the tests carried out are very simple, and no additional information is provided concerning previous works.
RESPONSE: Thank you for your observation. Consequently, we have remarked our previous work in the revised manuscript to highlight the findings of our current work. In previous approaches, we did not deeply study the oxidant effect in the template during the conductive polymer synthesis and as we have shown in the introduction, it is a main issue regarding the electrical properties of the conductive polymer. Our revised manuscript is a step forward for the synthesis point of view, since 48 hours soaking time has proven to be necessary for the optimal electrical properties in the final material and for the use of biomass-waste biopolymers which is good in a circular economy. We hope you may see the importance of our data for researchers involve in the field.
Reviewer 4 Report
The study deals with preparation and the analysis of physical properties of starch-based cryo-gels manufactured from biomass waste. The use of biomass waste for preparation of new polymer materials is important, but the area of applications of cryo-gels is not specified.
The authors referred to Ref. [16] for comparison of physical properties of cryo-gels from native corn starch and starch from biomass, but no details were provided. In particular, mechanical properties of the cryo-gels were not investigated. The question, whether the starch from bio-wastes can replace the conventional starch in applications remains unanswered.
Some inconsistencies arise in the interpretation of observations. On p. 2, the authors treated interactions between Fe ions and polymer chains as coordination bonds, whereas on p. 9, they referred to formation of ionic complexes to explain their findings.
The effect of soaking time on the Nyquist curves depicted in Fig. 6 seems interesting, but no quantitative explanation was provided for changes in their shapes.
I recommend to revise the concluding section. The last statement is unclear. The novelty of the work, its area of application and significance require clarification.
Author Response
Manuscript ID: polymers-2291667, Research Article.
Title: "Preparation of a conductive porous composite via iron(III) chemical oxidative deposition using on a biopolymer based -cryogel produced from biomass waste"
Authors: Laria Rodríguez-Quesada¹*, Karla Ramírez-Sánchez², Sebastián León-Carvajal1, Giovanni Sáenz-Arce3-4, Fabián Vásquez-Sancho5-6, Esteban Avendaño-Soto5-6, Juan José Montero-Rodríguez7 and Ricardo Starbird-Perez²
We are very grateful to all the reviewers for their helpful and thoughtful comments. We addressed all the concerns and modified the text accordingly. A detailed response to each comment is provided below:
Reviewer: 4
COMMENTS FOR THE AUTHORS
- The study deals with preparation and the analysis of physical properties of starch-based cryo-gels manufactured from biomass waste. The use of biomass waste for preparation of new polymer materials is important, but the area of applications of cryo-gels is not specified.
RESPONSE: Thank you very much for your recommendation. The details of the cryogel systems application proposed on this manuscript have been added to the introduction section and as conclusion.
Additional references were added to the document to support the potential use of functionalize porous materials: [Liu, Y., Liu, J., & Song, P. (2021). Sustainable Materials and Technologies, 27, e00240], [Alvarado-Hidalgo, F., Ramírez-Sánchez, K., & Starbird-Perez, R. Molecules, 25(22)], and [F. Quignard, F., Valentin, R., & Di Renzo, F. (2008). New journal of chemistry, 32(8), 1300-1310].
From line 45 to line 48: “From the technological perspective, the capacity to modulate the biomaterial properties to convey unique characteristics allows their application in different fields. Biopolymers are excellent raw material candidates for aerogel processing regarding circular economy principles, relying on renewable raw material or energy sources [4,9,10].”
- The authors referred to Ref. [16] for comparison of physical properties of cryo-gels from native corn starch and starch from biomass, but no details were provided. In particular, mechanical properties of the cryo-gels were not investigated. The question, whether the starch from bio-wastes can replace the conventional starch in applications remains unanswered.
RESPONSE: We thank you for your comment. However, we have demonstrated in our current and revised manuscript that the extracted starch was structured in a porous scaffold and our conductive composite has successfully replicated its morphology. Therefore, it is confirmed that the starch from bio-wastes can replace the conventional starch as template. In the reference [16], mechanical and electrical characterization were performed for a specific formulation. Currently, we are studying those properties (by complementary methodologies named dynamic mechanical analysis, dielectric characterization, and atomic force microscopy) on specific formulations (containing mainly starch). It is our goal, in a further research paper, to report the mechanical and dielectric properties of those formulated composites.
- Some inconsistencies arise in the interpretation of observations. On p. 2, the authors treated interactions between Fe ions and polymer chains as coordination bonds, whereas on p. 9, they referred to formation of ionic complexes to explain their findings.
RESPONSE: Thank you for your observation, the correct (and documented) interaction between starch and iron ions is the formation of coordination complex. We apologize for the mistake and it has been corrected.
- The effect of soaking time on the Nyquist curves depicted in Fig. 6 seems interesting, but no quantitative explanation was provided for changes in their shapes.
RESPONSE: We totally agree with you. Supplementary material (Table S4) presents the quantification of the EIS data, and it has been used on discussion section from line 324 to line 326: “The fitting data confirmed that longer soaking time led to higher capacitance for the first and second semicircles, which is expected for the conductive polymer deposition [18].”
- I recommend to revise the concluding section. The last statement is unclear. The novelty of the work, its area of application and significance require clarification.
RESPONSE: Thank you for your recommendation. We have improved the conclusion according to your observation, emphasizing the innovation of the manuscript.
From line 337 to line 349: “Our work gives insights in the methodology to functionalize starch obtained from biomass waste using chemical polymerization of the poly(3,4-ethylenedioxythiophene) directly on the substrate. Thermal and spectrometric characterization of those composites allows us to detect and quantify the iron-starch interaction and the amount of oxidizing agent along the soaking time. Moreover, we have confirmed using electronic microcopy and nitrogen physisorption that the porous template endured the polymerization process through all preparation stages without an evident collapse on the macrostructure, slightly affecting the microstructure due to the PEDOT deposition on the template surface. Longer soak time (i.e., 48 hours) unveiled, by impedance spectroscopy, a more complex electrical behavior, which is of great interest to a wide audience, including electronic, environmental, and biological applications. Finally, the conversion of potato waste into a conductive porous composite is an example of the potential preparation of innovative materials that balance progress and sustainability in a circular economy framework.”
Reviewer 5 Report
The method for sample preparation as well as characterization sections are thoroughly described which render the section results and discussion easily readable and convincing . This paper contains interesting and new experimental results that deserve to be published.
Author Response
Manuscript ID: polymers-2291667, Research Article.
Title: "Preparation of a conductive porous composite via iron(III) chemical oxidative deposition using on a biopolymer based -cryogel produced from biomass waste"
Authors: Laria Rodríguez-Quesada¹*, Karla Ramírez-Sánchez², Sebastián León-Carvajal1, Giovanni Sáenz-Arce3-4, Fabián Vásquez-Sancho5-6, Esteban Avendaño-Soto5-6, Juan José Montero-Rodríguez7 and Ricardo Starbird-Perez²
We are very grateful to all the reviewers for their helpful and thoughtful comments. We addressed all the concerns and modified the text accordingly. A detailed response to each comment is provided below:
Reviewer: 5
COMMENTS FOR THE AUTHORS
Reviewer: 5
COMMENTS FOR THE AUTHORS
- “The method for sample preparation as well as characterization sections are thoroughly described which render the section results and discussion easily readable and convincing . This paper contains interesting and new experimental results that deserve to be published.”
RESPONSE: Thank you very much for your response.
Round 2
Reviewer 1 Report
The revised version of the manuscript is good enough to be published. The authors addressed each comment in detail and made appropriate changes.
Author Response
Reply to reviewers:
Manuscript ID: polymers-2291667, Research Article.
Title: " Preparation of a conductive porous composite via iron(III) chemical oxidative deposition using on a biopolymer based -cryogel produced from biomass waste"
Authors: Laria Rodríguez-Quesada¹*, Karla Ramírez-Sánchez², Sebastián León-Carvajal1, Giovanni Sáenz-Arce3-4, Fabián Vásquez-Sancho5-6, Esteban Avendaño-Soto5-6, Juan José Montero-Rodríguez7 and Ricardo Starbird-Perez²
Reviewer 1.
- The revised version of the manuscript is good enough to be published. The authors addressed each comment in detail and made appropriate changes.
Response: Thank you very much. We have found your observations quite useful, and they enriched our current manuscript.
Reviewer 2 Report
The author addressed almost all the issues but have still required few minor changes before acceptance.
1. In the introduction section” cryogels are obtained by freeze drying hydrogels at low temperatures”, sentence change to negative or sub-zero temperature in the introduction part. Low temperature is wrong concept.
2. Why you not include TGA in the revised manuscript. If possible, include it in the final version.
3. For such type of complex study more characterization required like TEM and XPS too.
4. EDS and XRD should move to main part of the manuscript from supporting information.
Author Response
Reply to reviewers:
Manuscript ID: polymers-2291667, Research Article.
Title: " Preparation of a conductive porous composite via iron(III) chemical oxidative deposition using on a biopolymer based -cryogel produced from biomass waste"
Authors: Laria Rodríguez-Quesada¹*, Karla Ramírez-Sánchez², Sebastián León-Carvajal1, Giovanni Sáenz-Arce3-4, Fabián Vásquez-Sancho5-6, Esteban Avendaño-Soto5-6, Juan José Montero-Rodríguez7 and Ricardo Starbird-Perez²
Reviewer 2
- In the introduction section” cryogels are obtained by freeze drying hydrogels at low temperatures”,sentence change to negative or sub-zero temperature in the introduction part. Low temperature is wrong concept.
Response: We agreed with your observation. Thus, the text has been modified accordingly.
- Why you do not include TGA in the revised manuscript. If possible, include it in the final version.
Response: We do not modify either the TGA data or discussion. The current manuscript has been reviewed and the thermal data is still on it.
- For such type of complex study more characterization required like TEM and XPS too.
Response: We appreciated your suggestions. TEM is a powerful tool regarding nanostructures and their crystallinity. However, we have used SEM and BET to compare the template and final composite porosity properties. Regarding XPS, it can provide valuable information regarding PEDOT oxidation state, but it may depend on the charge carrier concentration as we have previously stated using Raman and EIS analysis [Hernandez-Suarez, P., Ramirez, K., Alvarado, F., Avendano, E., & Starbird, R. (2019). MRS Communications, 9(1), 218-223]. Based on your comment, we totally agree that XPS, RAMAN, DRX or EPR measurements could be used in a further investigation as complementary data to support EIS analysis for a specific formulation.
- EDS and XRD should move to main part of the manuscript from supporting information.
Response: we appreciated your suggestions. The data is currently in the figure 5 and figure 6 respectively. Regarding the XRDs data, it was measured again using a cobalt anode to remove fluorescence from the diffractograms and thus improve the signal/noise ratio. In the current measurements, Starch and PEDOT patterns were identified, and they were included in the discussion.
From line 170 to line 175: “The bulk material was packed on a sample holder and measured in the range of (2θ) 5°–55° with a cobalt (Co) anode and an Fe filter using a GaliPix detector with a 17.8 mm anti scatter slit. The operating conditions including step size, voltage, and current were adjusted to 0.007°, 40 kV, and 40 mA, respectively. Finally, the 2θ data was recalculated from Cobalt-Kα to Copper-Kα for analysis, the background was removed and each diffractogram was normalized by area.”
From line 293 to line 301: “XRD diffraction (fig. 6) analysis showed patterns for the starch template in all the samples [51]. After the PEDOT deposition, it is observed a peak profile associated to the conductive polymer that shows some shifting due to molecular structural rearrangements caused by changes in its doing level [52,53]. Additionally, small, broad, and overlayed peaks are observed in the XRD data, they could be related to amorphous iron compounds. Similar results were reported during the synthesis of PEDOT using FeCl3 as oxidant in absence of template [54]. Therefore, considering the iron and PEDOT tendency to saturate the template after 8 hours soak time, our FTIR and XDR results may indicate that iron ions differ in their coordination interactions with the starch template along the soak time.
Reviewer 3 Report
After reading the revised version and the answer of the authors, I consider the following revisions need to be addressed in the manuscript.
1) The introduction should be revised to discuss the importance of circular economy and the reusage of biomass waste to highlight the novelty that authors claim.
2) A schematic diagram that summarizes the process, including the chemical structure of the biomass employed and PEDOT polymerization should be added.
3) The derivate curves of TGA spectra needs to be added to determine the transition temperatures.
4) FTIR spectra in the area 1750-500 cm-1 should be zoom-in for a better visualization of the peaks of interest discussed in the text. The differences observed in the band located at around 1650 cm-1 need to be discussed.
5) The control image of the sample at time 0 before soaking in iron needs to be included in the Figure 4. In addition to this, this figure does not reflect the finding that authors mention in the text, “the ratio of sulfur in the sample decreased after 8 hours of soaking”, as all the images look like similar.
6) Concerning Figure 6, I suggest showing the spectrum of the sample at time 0, as well as grouping all the spectra for 0, 0.5, 8, 24, and 48h in one plot for a better comparison between samples with the same scale in the x- and y-axes. Besides, the CV curves should be also grouped altogether and shown in a separate plot. Thus, the Figure 6 would be divided in two, Figure 6a with the Nyquist plots and Figure 6b with the CV curves.
7) The references section needs to be double-checked to include the volume and the article pages or article number of each reference.
Author Response
Reply to reviewers:
Manuscript ID: polymers-2291667, Research Article.
Title: " Preparation of a conductive porous composite via iron(III) chemical oxidative deposition using on a biopolymer based -cryogel produced from biomass waste"
Authors: Laria Rodríguez-Quesada¹*, Karla Ramírez-Sánchez², Sebastián León-Carvajal1, Giovanni Sáenz-Arce3-4, Fabián Vásquez-Sancho5-6, Esteban Avendaño-Soto5-6, Juan José Montero-Rodríguez7 and Ricardo Starbird-Perez²
Reviewer 3
- The introduction should be revised to discuss the importance of circular economy and the reusage of biomass waste to highlight the novelty that authors claim.
Response: we appreciated your suggestions. We have included detailed information to support our claims.
From line 35 to line 48: “Natural polymers enable the production of innovative materials within the framework of the principles of the circular economy and it is a key strategy to balance progress and sustainability [1]. The circular economy concept is based on the material resources reuse, reducing the generation of waste, relying on efficient green technologies that reduce the use of hazardous chemicals, and promoting growth through innovation [2]. Biomass is a sustainable, green, non-toxic [3], biodegradable [4] and abundant natural resource [5], and a significant portion of lignocellulosic biomass waste is discarded from agricultural activities [6]. For instance, starch is a natural polymer obtained from potato agricultural biowastes [7,8]. The management of biomass waste is a growing challenge in developing countries due to its increasing generation and the lack of feasible and efficient approaches to provide proper use of these resources [9]. Their conversion into value-added products is possible through its structuration and functionalization [10]. Accordingly, our approach opens a novel use of the starch biomass to prepare added value composites using green technologies.”
- A schematic diagram that summarizes the process, including the chemical structure of the biomass employed and PEDOT polymerization should be added.
Response: we appreciated your suggestions. The figure 1 has been added to illustrate our process.
- The derivate curves of TGA spectra needs to be added to determine the transition temperatures.
Response: we appreciated your suggestion. The information has been added in the current figure 2.
- FTIR spectra in the area 1750-500 cm-1 should be zoom-in for a better visualization of the peaks of interest discussed in the text. The differences observed in the band located at around 1650 cm-1 need to be discussed.
Response: we appreciated your suggestion. The suggested data have been zoomed in in the figure 3, inset. And additional discussion on the band at 1650 cm-1 has been added to the manuscript.
From line 235 to line 236: “Shifting of this signal when the conductive polymer is deposited on the template, may be due to changes in the Starch/PEDOT interaction.”
- The control image of the sample at time 0 before soaking in iron needs to be included in the Figure 4. In addition to this, this figure does not reflect the finding that authors mention in the text, “the ratio of sulfur in the sample decreased after 8 hours of soaking”, as all the images look like similar.
Response: We appreciated your suggestion. The starch template data was not added in the original manuscript due to the total absence of iron and sulfur in its chemical composition. The information regarding the EDS data has been added in the figure S2. Additionally, no quantitative information has been extracted from EDS to evaluate the sulfur and iron content in the samples, they are only illustrative data. TOC and TGA are quantitative analysis that have been studied to address the sulfur and iron ration variation in the samples.
- Concerning Figure 6, I suggest showing the spectrum of the sample at time 0, as well as grouping all the spectra for 0, 0.5, 8, 24, and 48h in one plot for a better comparison between samples with the same scale in the x- and y-axes. Besides, the CV curves should be also grouped altogether and shown in a separate plot. Thus, the Figure 6 would be divided in two, Figure 6a with the Nyquist plots and Figure 6b with the CV curves.
Response: The template impedance and CV data has been measured and added in the fig S2. Starch is a non-conductive material; therefore, its electrical characterization data (e.g., CV and Z) only shows the equipment capabilities or limits, and no further analysis can be done from them, except to confirm its insulation sample behavior. In general, the CV data is kept as inset data since they only illustrate the conductive nature of the composite. Due to the sample porosity, Impedance data and Nyquist plots are more suitable to describe the electrical behavior of the composites.
- The references section needs to be double-checked to include the volume and the article pages or article number of each reference.
Response: We have checked the references to fulfill the references data requirements.
Round 3
Reviewer 3 Report
The article could be accepted for publication after minor revisions:
1) The plots included in the Figure 5 (vi) are not visible.
2) The insets of the Figure 8a-d are not visible, as well as the labels highlighting some points of the curves cannot be easily readable.
Author Response
Reply to reviewer:
Manuscript ID: polymers-2291667, Research Article.
Title: "Preparation of a conductive porous composite via iron(III) chemical oxidative deposition using on a biopolymer based -cryogel produced from biomass waste"
Authors: Laria Rodríguez-Quesada¹*, Karla Ramírez-Sánchez², Sebastián León-Carvajal1, Giovanni Sáenz-Arce3-4, Fabián Vásquez-Sancho5-6, Esteban Avendaño-Soto5-6, Juan José Montero-Rodríguez7 and Ricardo Starbird-Perez²
We are very grateful for their helpful and thoughtful comments. We have addressed your concerns and modified the figures accordingly. A detailed response to each comment is provided below:
Reviewer: 3
COMMENTS FOR THE AUTHORS
RESPONSE:
- “The plots included in the Figure 5 (vi) are not visible”
Response: We appreciate your observation. The figure was included at high resolution in the “Figures.zip”, however in the manuscript, previously, it was attached at low resolution to reduce the file size. Currently, the figures are in the text (.doc file) at high resolution. Additionally, the figures were generated according to the Polymer criteria, and they were zipped and attached in the Polymer´s website.
- “The insets of the Figure 8a-d are not visible, as well as the labels highlighting some points of the curves cannot be easily readable”
Response: Thank you for your observation. We have corrected the quality of the image and confirmed that they are satisfactory according to the Polymer criteria. Additionally, the figure was zipped and attached in the Polymer´s website. Finally, converting the word to PDF is reducing the quality of the figures, so we suggest reviewing the zipped images for quality control.